# Genetic and Acquired Heterotopic Ossification: A Translational Tale of Mice and Men

**DOI:** 10.3390/biomedicines8120611

**Published:** 2020-12-14

**Authors:** Serena Cappato, Riccardo Gamberale, Renata Bocciardi, Silvia Brunelli

**Affiliations:** 1Department of Neurosciences, Rehabilitation, Ophthalmology, Genetics, Maternal and Child Sciences (DINOGMI), University of Genova, 16132 Genova, Italy; serena.cappato@edu.unige.it; 2School of Medicine and Surgery, University of Milano-Bicocca, 20900 Milano, Italy; riccardo.gamberale@unimib.it (R.G.); silvia.brunelli@unimib.it (S.B.); 3UOC Genetica Medica, IRCCS Giannina Gaslini, 16147 Genova, Italy

**Keywords:** heterotopic ossification, mouse models, fibrodysplasia ossificans progressiva, FOP, ACVR1/Alk2, BMP signalling, Activin A, GNAS1, POH, SARS-CoV-2

## Abstract

Heterotopic ossification is defined as an aberrant formation of bone in extraskeletal soft tissue, for which both genetic and acquired conditions are known. This pathologic process may occur in many different sites such as the skin, subcutaneous tissue, skeletal muscle and fibrous tissue adjacent to joints, ligaments, walls of blood vessels, mesentery and other. The clinical spectrum of this disorder is wide: lesions may range from small foci of ossification to massive deposits of bone throughout the body, typical of the progressive genetically determined conditions such as fibrodysplasia ossificans progressiva, to mention one of the most severe and disabling forms. The ectopic bone formation may be regarded as a failed tissue repair process in response to a variety of triggers and evolving towards bone formation through a multistage differentiation program, with several steps common to different clinical presentations and distinctive features. In this review, we aim at providing a comprehensive view of the genetic and acquired heterotopic ossification disorders by detailing the clinical and molecular features underlying the different human conditions in comparison with the corresponding, currently available mouse models.

## 1. Introduction

Heterotopic ossification (HO) is a pathological process leading to the neoformation of mature bone in extraskeletal, soft tissues.

In physiological conditions, bone formation occurs during development through two main pathways: endochondral ossification, in which a cartilage intermediate is progressively replaced by osteogenic cells; and intramembranous ossification, where osteogenic cells differentiate from condensed mesenchymal cells [1,2].

Development of heterotopic bone shares striking molecular and histological features with normal bone development and can be distinguished from ectopic tissue calcification, a pathological event in which different processes lead to the deposition of insoluble calcium salts of variable composition outside bone and teeth, with the involvement of different soft tissues and organs (skin, kidney, lungs, vessels, etc.). Ectopic calcification may be observed in aging, or secondary to different pathological conditions in the presence of normal calcium and phosphorous values in the plasma (dystrophic calcification). Alternatively, it may be the result of conditions with altered plasma ion levels with precipitation of the exceeding salts (metastatic calcification) [3,4].

Usually, HO is not associated with any metabolic conditions and occurs in different, broad clinical settings.

The most common forms of HO are acquired and can occur as a complication of surgery (e.g., arthroplasty), fracture repair, in response to muscle and soft tissue trauma, severe burns, traumatic injury of brain and spinal cord. Moreover, post-traumatic HO is a common complication in combat injuries which severely prevents patients from recovering or adapting to prosthesis [5]. Genetic forms of HO are rare and include the fibrodysplasia ossificans progressiva (FOP), progressive osseous heteroplasia (POH) and other GNAS1-related conditions.

The clinical spectrum of this condition is wide, lesions range from small foci of ossification to massive deposits of bone throughout the body leading to cumulative ankylosis and severe and progressive impairment of movement [6]. This latter course is more typical of the genetic forms of HO, although also acquired lesions may be clinically relevant and relapses at the site of the lesion may occur in response to interventions aimed at removing the bone neoformation.

Regardless of the etiology, it is hypothesized that in the pathogenesis of HO the main factors that play a role are: (a) a genetic susceptibility (mendelian condition or predisposing genetic and multifactorial background); (b) an inciting event, such as different types of trauma able to function as a trigger, although not always recognizable; (c) the generation of a conductive environment at the site of injury through a complex crosstalk between the cells of the damaged tissue, inflammation and progenitor cells, that may be of multiple origin, able to differentiate into bone after receiving the appropriate signal at an available receptor.

This work provides an overview of the main forms of HO, from acquired forms to severe genetic conditions, among which fibrodysplasia ossificans progressiva, with a focus on the available mouse models mirroring the human counterparts (Table 1).

We have subdivided the human conditions into acquired and genetic forms with a description of the corresponding available mouse models. Nevertheless, this classification may be considered subjective to a certain extent, since it has become evident that injury and inflammation can accelerate and trigger HO in FOP patients as well as in animal models of genetic HO, and the high variability in predisposition of different individuals to acquired HO suggests a genetic basis for individual susceptibility.

## 2. Acquired Heterotopic Ossification

### 2.1. Post-Traumatic Heterotopic Ossification

The formation of new bone through an endochondral process is an uncommon event in the postnatal life initiated by different triggers such as fractures, traumatic events, acute trauma, combat related injuries and severe burns [7,8,78,79]. Moreover, HO is a frequent complication of orthopedic surgeries, involving hip and elbow [5,9,10,11].

The frequency of this secondary event is variable and seems to be strongly correlated with the site of trauma, the severity of the insult and also with the patient’s age [8,80].

An effective therapeutic approach to prevent and treat ectopic bone formation is crucial for a positive resolution of interventions and quality of patients’ life but is still not available. Up to now, the three possible therapeutic options consist of surgical excision (although relapses maybe common), treatment with anti-inflammatories and radiation therapy [7,12] with frequent limitations related to the accessibility of the involved areas.

The study of different types of nonhereditary ectopic bone specimens has revealed that osteogenic differentiation in damaged tissues is the result of lymphocytic recruitment and migration, followed by fibroproliferation and vascularization that leads to mature bone formation through a cartilage intermediate [13]. The involvement of the innate immune system is certainly crucial although the exact role and balancing of the different components still need to be better clarified.

The understanding of the HO core process including the strong relationship between the inflammation, pro-osteogenic stimuli and precursor cells requires the elements and pathways that contribute to the activation of the endochondral differentiation leading to ectopic bone formation to be defined [25].

#### Mouse Models of Post-Traumatic Heterotopic Ossification HO

The etiology of acquired HO still remains unclear as there are many factors contributing to its development, including inflammation, hypercalcemia, hypoxia and immobilization [81]. Since little is known about the underlying causes and the pathophysiological mechanisms of acquired HO, it has become difficult to develop new mechanism-based animal models. Some uncertainties also remain on their accuracy in reproducing human features of HO [14]. Anyway, nowadays there are many animal models able to reproduce at least some of the features of typical HO.

*Bone morphogenetic protein (BMP) injection/implantation models.* BMPs are signalling molecules belonging to the family of the transforming growth factor β (TGF-β) involved in the homeostasis and differentiation of a wide range of tissues, including cartilage and bone [15,82]. In the context of HO, several studies have been performed to better understand the role of BMPs in the induction of osteogenesis. 

Some BMPs, such as BMP2 and BMP4, are able to induce potent inflammatory reactions and their injection, with or without additional injury, is followed by a robust endochondral ossification process [16,83]. In particular, a BMP2 injection, together with mild cardiotoxin-mediated muscle injury, elicits infiltration of CD11b macrophages and endochondral ossification in less than 10 days [17,84].

In contrast, other BMPs, such as BMP9, are able to induce a weak inflammatory response and require other inflammatory stimuli to trigger heterotopic ossification [85]. To ameliorate the efficacy of the induction protocol, new strategies were developed for the delivery of BMPs in the muscles, including the implantation of BMP-loaded biomaterials, such as matrigels and sponges [86]. These strategies allow a slow, but constant, release of BMPs avoiding an impairment of muscular function at the same time. 

Another interesting approach comes from microporous calcium phosphate ceramic particles. These molecules do not release BMPs, but their implantation is still able to induce HO since these biomaterials improve the adhesion, proliferation and differentiation of cells, showing an improved osteoinductive ability [87]. However, the biological mechanisms underlying this process are not entirely understood, even if some theories assume that their physicochemical and structural characteristics may play a key role in this context.

Still, there are some debates concerning the nature of such models, since large amounts of BMPs are suddenly introduced into the muscle, giving rise to a nonphysiological HO development [14]. However, implantation models are frequently used since they can reproduce in a reasonable way the features of human HO, both from a molecular and a histological point of view. Furthermore, these models allow a local activation of the BMP signalling in the tissue, thus avoiding systemic effects. BMP4 overexpression has been observed in lesions of FOP patients suggesting that implantation models could also be appealing for the study of other forms of HO including the genetic forms [16].

These implantation models have been therefore particularly useful in investigating the process of ectopic ossification, which cells are involved in the onset and progression of HO [18,61,84] and are also amongst the most straightforward in vivo models to test new pharmacological approaches to inhibit HO [17,88]. 

*Achilles tenotomy model.* The ossification of the Achilles tendon is a rare event that can occur as a consequence of trauma or surgery. [89,90,91]. This process was investigated in animal models in which it was possible to induce HO following a trauma, such as tendon squeezing or dissection [92,93]. Some works have demonstrated that one of the key events underlying ectopic bone formation in the Achilles tenotomy models is hypoxia. A low oxygen tension environment is translated into an increase of hypoxia inducible factor (HIF)-1α expression and in turn, HIF-1α enhances angiogenesis, chondrogenesis and finally osteogenesis [94]. Nowadays, Achilles tenotomy models are sometimes used due to their straightforwardness and reproducibility [81] even though they are still of doubtful relevance for humans, since ectopic bone formation in the Achilles tendon is a rare event, often associated with other pathologic conditions such as rheumatoid arthritis and ankylosing spondylitis [93].

*Burn/tenotomy model.* A further evolution of the tenotomy model is represented by the burn/tenotomy model. In humans, burn injuries are traumatic events commonly causing HO, even though it is hard to predict where the ossification will occur [95]. The burn/tenotomy model involves the combination of tendon dissection and burn injury on the dorsal skin, which is required in order to elicit a systemic inflammatory state [19]. This mouse model develops HO in the areas that received the dissection with high frequency, such as the calcaneus, ankle joint, and tibia/fibula of the limb. As observed for the Achilles tenotomy model, mice receiving the burn/tenotomy treatment show an increase in HIF-1α expression, which plays a pivotal role in the development of traumatic HO [96]. Here as well HIF-1α supports angiogenesis, by upregulating VEGFA in the injured area, creating a proper microenvironment for subsequent endochondral ossification [97]. This model presents some advantages: in fact, by combining tenotomy and burn injury, the ossification sites are more predictable and HO develops in an accelerated way. In addition, HO induction does not require the administration of exogenous molecules [19,96,97,98]. Nevertheless, both BMP and TGF-β signalling pathways are upregulated after the burn/trauma, but with a difference to the genetic FOP model (see below), Activin A does not appear to play a role in the ectopic ossification [99].

The burn/tenotomy model has been widely used to investigate which cells are involved in the onset and progression of HO after trauma, highlighting the critical role of several types of circulating mesenchymal cells and cells of the innate immune system [20,21,100,101]. 

*Michelsson’s model.* This model is also known as the “immobilization−manipulation model” and was first ideated by Michelsson who was able to induce HO in rabbit quadriceps by repeated and intense immobilization of the knee joint, which could be similarly reproduced in other joints [22]. This model turns out to be particularly useful to understand the role of inflammation in HO, in particular an increase in the level of prostaglandins has been observed before proper bone formation [23]. Moreover, it has been shown that the formation of HO can be prevented by separating the femur and the quadriceps with the insertion of a plastic membrane. The interaction between bone and muscle appears to be pivotal for the onset of HO. 

However, this model has received some criticism for what concerns the development of HO, since it is unclear whether the newly formed bone could be considered as ectopic bone or as a dystrophic calcification [24]. Furthermore, although Michelsson’s model paved the way for the study of HO in rabbits, its relevance in the context of other mammals, like mice, remains to be investigated.

### 2.2. Neurogenic Heterotopic Ossification

Neurogenic heterotopic ossification (NHO), affecting soft/extraskeletal tissue surrounding hip, shoulder and elbow joints, is a spontaneous consequence of injuries involving the central nervous system.

The primary damage leading to HO can directly involve traumatic brain injury (TBI) and spinal cord injury (SCI) but can also be a consequence of isolated nontraumatic neurological events such as stroke and cerebral anoxia [26,27,28]. 

The etiology and the severity of the primary neurological damage, the patient’s post trauma management, coma, ventilation support, autonomic dysregulation, spasticity and the gap between trauma and rehabilitation are all factors that can strongly contribute to the risk of ectopic bone formation and influence the HO locations and volumes [26,27]. 

The pathogenic mechanism of NHO is not yet well understood. The study of human lesions reveals some important points to be further investigated, such as the contribution of neuro-inflammation signals (e.g., substance P, calcitonin gene-related protein, CGRP, etc.) deriving from the damaged peripheral or central nervous system and the responsiveness of different resident precursor cells that activate the wrong repair process [25,29,30,31]. 

Very recently, Meyer and colleagues described four patients with a severe form of SARS-CoV-2 infection (COVID-19) developing HO of the hips and shoulders. All the patients required intensive care, with mechanical ventilation and a prolonged immobilization period [32]. The relationship between HO and COVID-19 is not clear. However, SARS-CoV-2 infection induces a potent systemic inflammation state, triggers macrophage activity and the production of inflammatory cytokines at tissue level, defined as a “cytokine storm” [102]. Mechanical ventilation may affect acid-base homeostasis thus inducing hypoxia. Moreover, severe infection spreads to the central and peripheral nervous system with high risk of encephalitis, stroke, and severe neuro-muscular illness [33].

These events, together with the prolonged immobilization of the patients, are all critical factors able to drive HO formation in COVID-19 patients. As commented by Meyer and coll., occurrence of this complication may be currently underestimated in severely affected patients and might further impact their rehabilitation.

#### Mouse Models of NHO

*Spinal cord injury (SCI) mouse model.* NHO is a frequent event and occurs in about 20 to 30% of patients following spinal cord injury [34]. From here, the necessity arose of developing a mouse model aimed at better understanding the features of ossification after SCI. In these models, this form of trauma is usually simulated by either a laminectomy of the dorsal spine, followed by a transection of the spinal cord and muscle injury by cardiotoxin injection [35] or by injury induction with a weight drop followed by the injection of a small dose of BMP2 [34]. Both models reproduce what is observed in patients with SCI that develop NHO, since ossification forms rapidly. Moreover, mice with SCI-induced NHO mirror the development of ectopic bone from the histological point of view, presenting a formation of lamellar bone with large amounts of osteoblasts, osteoclasts and osteocytes. However, since the procedure induces paraplegia of the mice, special care is needed to ensure the survival of the experimental animals, therefore this model may have a more limited use for large scale study.

The role of inflammation has been investigated also in NHO development. In particular, resident macrophages have been shown to produce several factors critical to the maturation and maintenance of newly formed bone, such as BMPs and Oncostatin M [31,35]. Therefore, macrophages may be another therapeutic target for the treatment of NHO.

*Traumatic brain injury (TBI) model.* NHO has been investigated in rats, with an attempt to create a model that could allow heterotopic ossification to be studied in association with TBI, coupled with other forms of peripheral injuries commonly observed in patients. In particular, these injuries consist of femoral fracture and muscle injury. It has been observed that after 6 weeks, 70% of the rats that received both forms of injuries, together with TBI induction, showed ectopic bone in the injured hindlimb [36]. Interestingly, only 20% of rats receiving both femoral fracture and muscle injury without TBI induction presented ectopic bone. For what concerned the ossification, joints showed the presence of ectopic bone as observed in human NHO patients; furthermore, it appeared to be more severe in rats in which TBI was combined with the other two forms of injuries compared to rats in which TBI was not induced [36]. Curiously, TBI has been proven to have a negative effect on bone healing in a rat model, while callus formation was exacerbated, probably as a consequence of the activation of different metabolic and inflammatory pathways [37].

Anyway, there are still some limitations concerning this model of NHO because of the lack of an assessment of the contribution of each individual injury to the development of ectopic bone formation, and of the clues concerning the histological analysis of the newly formed ectopic bone in tissues. Further studies may allow better clarification of these points.

## 3. Genetic forms of Heterotopic Ossification

Heterotopic ossification may also represent the most relevant clinical feature of three genetic diseases, fibrodysplasia ossificans progressiva (FOP), progressive osseous heteroplasia (POH) and Albright hereditary osteodystrophy (AHO). These are all rare conditions, inherited as autosomal dominant traits and characterized by the occurrence of bone neoformation in extraskeletal tissues. Nevertheless, these diseases significantly differ in the underlying genetic causes and pathways involved, clinical presentation and course, and in the differentiation process leading to the ectopic bone formation.

### 3.1. Fibrodysplasia Ossificans Progressiva (FOP)

Fibrodysplasia ossificans progressiva (FOP, OMIM135100) is a rare genetic disorder with an estimated average prevalence of 1–1.5/2,000,000 and one of the most severe conditions of HO.

The typical clinical presentation of FOP is characterized by the presence of a peculiar congenital malformation of the great toes that could be considered the first clinical sign of the disease, although other congenital anomalies (malformation of the thumbs, fusion of cervical vertebrae, digit reduction defects, etc.) and clinical signs (presence of tibial osteochondromas) may be present with variable expression and frequency [38]. HO of soft tissues, such as skeletal muscles, tendons, ligaments and joints starts in childhood and progresses throughout the life evolving to entrap patients in a second skeleton.

HO usually occurs with an episodic course consisting of acute phases called flare-ups alternating with quiescent phases of the disease activity. The study of the natural history of FOP reveals that flare-ups are preceded in more than 80% of the analyzed patients by symptoms like swelling, pain, or decreased mobility [38,39]. However, FOP progression can be extremely variable and unpredictable, not all the flare-ups may result in ectopic bone formation. On the other hand, HO may progress also with a creeping course, in the absence of a clinically relevant acute phase [38,39]. HO may be initiated or exacerbated by several factors such as trauma, vaccinations, surgical or medical interventions, infections, or may initiate without a recognizable trigger [38,39]. As such, early diagnosis of FOP is mandatory to prevent behaviors or procedures that might be harmful for the patient.

All these observations have suggested the importance of inflammation and immune response in the etiology of the disease. This is further supported by the histological studies performed on human specimens of biopsies obtained from patients before the diagnosis of FOP. In early lesions, the degeneration of the damaged tissue is evident and elicits a strong inflammatory response with tissue infiltration by different types of immune cells (monocytes, macrophages, lymphocytes, mast cells) [40,41]. Then, after a fibroproliferative phase, ectopic bone forms through a classical endochondral ossification process. This latter is further sustained by the markedly hypoxic microenvironment, generated by inflammation in the early FOP lesions, which enhances the BMP signalling and promotes HO formation [42]. The heterotopic bone has the features of a mature trabecular bone with marrow elements, with the same mechanical, physical and metabolic properties of the orthotopic bone.

The genetic cause of FOP is a gain-of-function mutation of the *ACVR1/Alk2* gene. The gene encodes a type I receptor for bone morphogenetic proteins (BMPs) [43], a wide group of secreted factors belonging to the TGF-β family of proteins. ACVR1/Alk-2 forms functional complexes at the cell membrane with type II receptors able to bind BMP ligands, thus activating both Smad-dependent and independent intracellular signalling pathways involved in osteogenesis and bone homeostasis [103].

The R206H is the most commonly recurrent mutation in FOP, affecting a highly conserved residue within the GS domain of the protein [38,43], whereas rare cases may be associated with different variants affecting the same functional region of the receptor or the kinase domain [71,72,73,74,75].

The mutation causes constitutive activation of the receptor which becomes hypersensitive to BMPs and, most importantly acquires a new, disease-specific feature by perceiving Activin A (ActA) as an agonist [49,50]. Activin A belongs to the same family of BMP ligands. However, usually it does not show osteogenic properties and although able to bind wild-type ACVR1/Alk2, in normal conditions this represents a non-transducing/inhibitory complex [49,50]. In contrast, binding of ActA to the mutated receptor carrying FOP-associated variants triggers the downstream Smad1/5/9 signalling [49,50,51], enhances the endochondral ossification of primary connective tissue progenitor cells of FOP patients [52], thus promoting HO formation.

Noteworthy, different types of immune cells (macrophages, dendritic cells, T and B lymphocytes, natural killer cells) are able to secrete and to respond to ActA, with a broad range of different modulatory actions on the inflammation process [53].

The different mouse models of FOP currently available are providing crucial insight into the role of ActA and the signalling pathways involved, the origin and nature of the different progenitor cells that contribute to the ossifying lesions, the role of inflammation and the importance of the microenvironment (hypoxia, etc.), and provide the basis to preclinical studies to develop targeted therapies.

#### Mouse Models of FOP

With the aim to reproduce the condition of the dysregulated BMP signalling occurring in FOP, several strategies have been adopted in mice.

*BMP ligand overexpression mouse models.* The first genetic strategy of mimicking FOP in vivo was to overexpress the BMP proteins involved in HO. BMP4 was highlighted as a key factor in FOP pathogenesis, therefore dysregulation of its expression was investigated in the FOP pathogenic context. The development of a model of BMP4 overexpression required the identification of a proper promoter that could drive its expression efficiently. Several promoters were investigated, but most of them were not able to induce postnatal HO or led to the onset of developmental abnormalities [104,105]. The only promoter that could induce the overexpression of BMP4, thus leading to proper HO formation was the neuron specific enolase (Nse) promoter [54].

Before the development of the *ACVR1/Alk2* mutated transgenic mice (see below), the Nse-BMP4 transgenic mouse has been the most used model for studying BMP overexpression in FOP. Nse-BMP4 mice mirror in a fair way the progressive formation of heterotopic bone seen in FOP patients and, as in humans, some sites like the diaphragm, tongue and extraocular muscles are spared from HO development. However, no malformations in the great toe and in the joints were observed, which are typical of FOP [54]. This model has been used for understanding which cell types can differentiate in the osteogenic lineage and for studying the events that trigger HO [54].

Interestingly the progeny deriving from the mating of Nse-BMP4 mice with mice overexpressing Noggin, an inhibitor of BMP4, do not develop FOP. Moreover, local injection of Noggin in a mouse model of BMP4-induced HO rescues the animals from developing heterotopic ossification in the site of injection, showing that the use of BMP inhibitors may be effective for the treatment of HO-related diseases [106]. 

*Hyperactive ACVR1/Alk2 models.* One of the first mouse models used to study FOP was actually generated to investigate the role of *ACVR1/Alk2* during development. This model was obtained by the expression of a Cre-inducible transgene consisting of the human *ACVR1/Alk2* cDNA carrying the engineered Q207D substitution (also known as constitutively active Alk2, caAlk2) [107].

This mutation causes the substitution of a glutamine with a negatively charged residue, namely aspartic acid, in the GS domain of the receptor, leading to constitutive activation of the downstream Smad-dependent cascade [107].

After the discovery of *ACVR1/Alk2* as the causative gene in FOP, this mouse was considered useful to model the disease phenotype, since intramuscular expression of the *caAlk2* transgene was able to induce ectopic endochondral bone formation with joint fusion and functional impairment [55].

In this model, global postnatal expression of *ACVR1/Alk2^Q207D^* obtained by mating the mice with ubiquitously expressed inducible Cre (CAGGCre^ERT^) did not develop HO. HO was observed when *ACVR1^Q207D^* mice were injected at specific sites with adenoviral vectors containing the Cre recombinase. Curiously, when mice with the global activation of the mutation were injected with control adenovirus, HO was developed as well [55].

These results led to the hypothesis that HO formation was dependent on the presence of both the *ACVR1^Q207D^* and an inflammatory trigger/environment [55,108].

The engineered *ACVR1^Q207D^* mutation has never been described in humans in association with FOP. However, the substitution of the same residue by a glutamic acid, Q207E, has been reported in rare cases of FOP [38]). Although Q207D and Q207E may look similar since they both introduce a negatively charged residue, these mutations have different impacts on the receptor function. In fact, ACVR1^Q207D^ was shown to be constitutively activated by an irreversible loss of inhibitory GS domain conformation occurring upon the first phosphorylation event, which is not observed in ACVR1^Q207E^ and ACVR1^R206H^ [56]. Although ACVR1^Q207D^ shows some functional features different from the naturally occurring ACVR1^Q207E^ and ACVR1^R206H^ mutants, the ACVR1^Q207D^ mouse model presents a robust, BMP-signalling dependent HO formation and is extensively used in the preclinical development of inhibitory compounds and drugs [55,57].

*Acvr1^R206H^ mouse models.* As soon as the FOP mutations were identified, great effort was put in place to produce a more disease-relevant animal model. As mentioned above, the great majority of FOP patients carry the same R206H mutation. The first models were obtained by introducing the *Acvr1^R206H^* mutated gene in the murine endogenous locus [58]. Even though the endogenous mutation led to the development of classic FOP features, like digit malformation, joint fusion and other skeletal anomalies, most of the progeny encountered problems of perinatal lethality. This allowed only the chimeric mice with estimated 70% to 90% mutated cells to be studied, but still the problem concerning perinatal lethality limited severely the applicability of this model [58]. For this reason, new strategies had to be explored.

In order to overcome the perinatal lethality, a model of conditional knock-in mutation was developed *Acvr1^[R206H]/FlEx^* [50]. When *Acvr1^R206H^* expression is induced postnatally upon tamoxifen-inducible Cre-mediated recombinase, HO is triggered and develops between 2 and 4 weeks, apparently without the need of additional injury. This model provided some important insights concerning the molecular mechanisms of FOP and made possible the investigation of the aberrant role of ActA in FOP. As previously stated, ActA has an inhibiting activity towards ACVR1/Alk2 in a wild-type background, but in patients and mice presenting the R206H mutation, ActA ends up being a powerful activator of the receptor, thus inducing HO [50]. The same floxed transgenic line has been used to develop another *Acvr1^R206H^* mouse model [59]. In this work upon doxycycline-induced Cre recombination and cardiotoxin-mediated muscle injury, complete HO developed in 2 weeks. Using this model, the immune system has been shown to play a pivotal role in FOP heterotopic bone formation. Cell types such as macrophages, mast cells and neutrophils have been observed in FOP lesions of *Acvr1^R206H^* mice after injury [59]. Moreover, these cells persisted at high levels during bone formation, instead of returning to preinjury levels, and increased production and persistence of proinflammatory cytokines, such as TNF-α, IL-6, and IL-1b, further strengthen the hypothesis of a sustained proinflammatory environment in FOP lesions of *Acvr1^R206H^* mice [59]. Depletion of mast cells and/or macrophages has been proven to reduce HO in *Acvr1^R206H^* mice, indicating that these cells may be candidate targets for pharmacological treatments in FOP [59]. However, a better clarification concerning the cell composition of the inflammatory infiltrate in FOP lesions is still needed. 

This model, as well as a different *Acvr1^R206H^-*knock-in floxed strain, in which expression of *Acvr1^R206H^* is Cre-dependent and under the control of the endogenous *Acvr1*
*locus* (*Acvr1^tnR206H^)*, that has been generated independently [60], have been used to characterize the cells that can contribute to the endochondral ossification, in particular fibroadipogenic precursors (FAP) [60,109].

### 3.2. Progressive Osseous Heteroplasia (POH) and GNAS1 Related Conditions

Progressive osseous heteroplasia (POH, OMIM 166350) is an ultrarare genetic disease that begins in early childhood with widespread heterotopic ossifications at dermal and subcutaneous fat level, and progresses with the involvement of subcutaneous and deep connective tissues [61,62,63]. The disease is mainly sporadic but recurrence with an autosomal dominant inheritance has been also reported [64]. The genetic causes of the disease are loss of function mutations of the Gs-α isoform of the of the *GNAS1* gene, in the inherited paternal allele [63,65,66,67].

The early manifestation of the disease is a maculopapular rash caused by patchy areas of bone within the dermis, present at birth or appearing some weeks later. Then, HO progresses from the skin and subcutaneous fat to deep connective tissues (subcutaneous fat, muscles, tendons, ligaments, fascia) by severely impairing joint mobility and limb growth. HO associated with POH is not triggered by trauma, infections, nor associated with metabolic abnormalities, and develops through an intramembranous differentiation process [61,63,65,68].

The *GNAS1* locus is characterized by a complex epigenetic regulation with the synthesis of different transcripts with mono and biallelic expression. As such, besides POH both constitutive and somatic mutations in the *GNAS1* gene with the differential involvement of the maternal or paternal alleles, result in a broad spectrum of phenotypes that may include HO, and a variety of clinical signs such ad skeletal malformations, hormone alterations and obesity. This group of diseases are Albright hereditary osteodystrophy, (AHO) pseudopseudohypoparathyroidism (PPHP) and different types of pseudohypoparathyroidism (PHP) [70,71].

In the context of heterotopic ossification, patients affected by AHO share a constellation of clinical manifestations including short stature, brachydactyly, obesity and ossifications limited to the subcutaneous layer (subcutaneous ossification, SCO) that could be considered the peculiar characteristics of this disorder. SCO occurs spontaneously or secondary to trauma, can cause pain and affect daily life quality and surgical removal does not guarantee a definitive resolution [63,72,73]. 

*GNAS1* encodes the stimulatory alpha subunit (Gαs) of the G protein complex. This latter transduces extracellular signals received by transmembrane receptors called G protein-coupled receptors (GPCRs) to cellular mediator by stimulating the activity of the hormone-sensitive adenylyl cyclase. Each G protein is a heterotrimer composed of an α, β, and γ subunit. Gαs-mediated signalling interacts with the Wnt and Hedgehog pathways, both crucial regulators of skeletal development, remodeling and injury repair [110].

Moreover, GNAS1 has a crucial role in skeletal development and homeostasis by regulating different processes of skeletal cell maturation. In 2011, Pignolo et al. observed that the altered GNAS1 expression promoted the osteoblast differentiation by unbalancing the differentiation of the multipotent connective tissue progenitor cells towards osteogenesis at the expense of adipogenesis [111]. Furthermore, the central role of Gαs has been demonstrated in the correct formation of skeleton bone by inhibiting/limiting Hedgehog (Hh) signalling in mesenchymal progenitor cells. Loss of function mutations in the *GNAS1* gene leads to the upregulation of Hh that is considered sufficient to induce HO in GNAS1-related conditions [110,112,113].

#### Mouse Models of POH and AHO

*Mouse models of POH.* The rCre-Gsα mouse model is a transgenic murine model expressing the Cre recombinase under the control of human renin (hRen) promoter, which can excise the *GNAS1* gene when flanked by loxP sites [69]. Unexpectedly, this kind of mutation had no major effects on the renin-angiotensin system and the urinary concentrating ability of rCre-Gsα mice was preserved [69]. Interestingly, mutated mice show marked abnormalities in the spleen due to fibrous connective tissue deposition, which are not found in human POH patients. 

On the other hand, this model reproduces some of the common features of human POH, in particular soft tissue mineralization and ossification, which may also extend to subdermal connective tissues [69]. Furthermore, ossification has been found also in the skeletal muscles adjacent to the long bones of the forelimb, which is another common site of ossification observed in human patients [62]. Surprisingly, the rCre-Gsα mouse model reproduces well most of the common features observed in human POH patients, showing that Gsα has a fundamental role in mineralization and bone development. Still, deeper studies concerning spleen fibrosis observed in this model may be needed, in order to prevent undesired effects in mice.

*Mouse models of AHO***.** Targeting the *GNAS1* gene has been the most direct strategy to mimic AHO in mice models. The first genetic approach was performed by targeting the exon 2 of the *GNAS1* gene, whose homozygous deletion is associated with postnatal lethality [74]. Different phenotypes were observed in these mice depending on the maternal or paternal origin of the allele. The animals with the maternal inherited mutation presented resistance to PTH, were obese and hypometabolic, whereas the paternal origin of the mutation was translated into lean and hypermetabolic mice [75]. The deletion of the exon 2 in the chondrocyte lineage, gave rise to ectopic cartilage formation in the growth plate area of the tibia, showing that *GNAS1* is a negative regulator of chondrocyte differentiation [114]. Still, no traces of ectopic bone were observed in these models. In this regard, targeting the exon 1 of the *GNAS1* gene turned out to be a more successful strategy [76]. In this murine model was observed the presence of subcutaneous ossification by 12 months of life, a typical feature of AHO affected patients. Furthermore, no differences concerning the maternal or paternal origin of the allele were observed, both in terms of ossification frequency and histological appearance [76]. On the other hand, male mice had more severe and widespread ossification in the subcutaneous tissues, indicating that probably androgens may accelerate the ossification process. Other studies also showed that the deletion of the exon 1 of the *GNAS1* gene was related to a decrease in sensitivity to PTH and TSH, with increased circulating levels of these hormones, with more severe phenotypes associated to the maternal origin of the mutation [77].

## 4. Conclusions 

Heterotopic ossification represents a pathological process that may occur in a broad spectrum of clinical presentation, as an isolated/acquired sign or as a feature of a genetic condition, from small and self-limiting lesions to progressive forms that cause severe disability.

In this work, we have summarized the different presentations of HO in humans, with attention to both acquired and genetic forms such as FOP. Most importantly, we have provided a systematic comparison between the human condition and the corresponding animal model (Table 1), highlighting the adherence and differences with the human counterpart thus underlining the strengths and the critical points of each.

The availability of a condition-relevant animal model is of critical importance: to clarify in detail the molecular and cellular mechanisms featuring the progression of the disease and to provide preclinical evaluation of promising therapeutic agents.

## Figures and Tables

**Table 1 biomedicines-08-00611-t001:** Overview of the main forms of acquired and genetic Heterotopic Ossification (HO) in humans and comparison with the corresponding available mouse models.

HO Classification	Human Condition		Mouse Models of HO	
**Acquired HO**	**Inciting Event/Condition**	**Features**	**Inciting Event**	**Features**
**Post-traumatic HO**	Fracture Trauma Combat related injuries Severe burns Arthroplasty/Surgery [5,7,8,9,10,11,12,13]	Correlation with the site of trauma and the severity of the injury Strong inflammatory response	BMPs injection /implantation models [14,15,16,17,18]	Robust HO No BMP-related systemic effects HO may vary depending on the type of BMP applied Applicable for genetic forms of HO Nonphysiological HO development
			Achilles tenotomy model Burn/tenotomy model [19,20,21]	HO in the injured hindlimbs High inflammatory response HIF-1α increased expression
			Michelsson’s model [22,23,24]	HO as a consequence of bone−muscle interaction Inflammatory infiltrate increases prostaglandin levels Unclear HO entity Mostly used in rabbits
**Neurogenic HO**	Spinal cord injury Traumatic brain injury Encephalitis Stroke Severe myopathy and neuropathy Prolonged immobilization [25,26,27,28,29,30,31,32,33]	Correlation with the severity of the injury Post trauma management complications COVID-19 related complications	Spinal cord injury (SCI) mouse model [31,34,35]	Rapid neurogenic heterotopic ossification (NHO) Well-reproduced NHO histology NHO requires additional inflammatory stimuli Paraplegic mice Animal survival may be affected
			Traumatic brain injury (TBI) mouse model [36,37]	NHO in joints and in the injured femurs NHO severity increases with multiple injuries No exogenous molecules required for NHO
**Genetic forms of HO**	**Genetic cause**	**Features**	**Genetic background**	**Features**
**FOP** (OMIM 135100)	Gain of function mutations of *ACVR1/Alk2*: alteration of the BMP signalling and acquired responsivity to Activin A [38,39,40,41,42,43,44,45,46,47,48,49,50,51,52,53]	Congenital malformation of the great toes Variable association with other skeletal anomalies (thumb and digit malformation, fusion of cervical vertebrae, etc.) Development of tibial osteochondromas Endochondral HO Spontaneous and trauma-induced HO Flare-ups with preosseous swelling and inflammation Involvement of skeletal muscles, tendons, aponeuroses, fascia Progressive and severely disabling	BMP ligand overexpression mouse models (Nse-BMP4) [54]	FOP-like phenotype Progressive endochondral ossification No great toes malformation Inflammatory infiltrate in lesions
			Alk2*^Q207D-floxed^* (*caAlk2^fl^*) mouse model [55,56,57]	Postnatal and progressive HO after Cre induction Different response to BMP inhibitors Different ACVR1 conformation
			*Acvr1^R206H/+^* chimeric mouse model [58]	Malformed first digits and hindlimbs Postnatal and progressive HO Limited and impaired mobility Postnatal lethality Only progeny with 70–90% mutated cells available for analysis
			*Acvr1^[R206H]FlEx^* knock-in mouse model [50,59] *Acvr1^tnR206H^* knock-in mouse model [60]	Postnatal and progressive HO after Cre induction Altered Activin A signalling Inflammatory infiltrate No great toes malformation No postnatal lethality
**POH** (OMIM166350)	Loss of function mutation of *GNAS1* (paternal allele) [61,62,63,64,65,66,67,68]	Maculopapular rash HO of the skin and dermis mainly by intramembranous differentiation process Progressive involvement of subcutaneous and deep connective tissues Progressive may be severely disabling	rCre-Gsα mouse model [69]	HO in soft dermal tissues HO is invasive Ossification in skeletal muscles adjacent to the long bones Fibrotic spleen
**AHO** (OMIM103580 & 612463)	Loss of function mutation of *GNAS1* (mainly maternal transmitted) [63,70,71,72,73]	Short stature Brachydactyly Obesity Subcutaneous ossifications Other skeletal anomalies +/− endocrine abnormalities such as multihormone resistance *	*Gnas^E2+/−^* mouse model [74,75]	Maternal mutation PTH resistance Obesity Hypometabolism Paternal mutation Lean mice Hypermetabolism No HO formation
			*Gnas^E1+/−^* mouse model [75,76,77]	Decreased sensitivity to PTH and TSH Increased circulating PTH and TSH Subcutaneous HO

*,“+, presence” and “−
, absence”.

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
