# Peer review of "Genetic and Acquired Heterotopic Ossification: A Translational Tale of Mice and Men"

_biomedicines, 2020, doi:10.3390/biomedicines8120611_

Round 1
Reviewer 1 Report
This review updated the current progress on heterotopic ossification(HO). The author provided a systematic comparison between human disease and animal models.
The author should cite the papers with hypoxia (PMID: 27027798) and activin A (PMID: 29170109) in genetic HO of FOP.
In line 328,"ACVR1Q207D model may not represent the most suitable model to be used in preclinical studies of potential pharmacological treatments for FOP" This model had robust HO than R206H model, shared the same BMP signaling pathway, and used widely for developing some drug candidate. The conclusion is too arbitrary. Suggest to revise the tone.
Interestingly, the author updated a new report of few cases of COVID-19 who developed HO and give a reasonable analysis.
Author Response
Response to Reviewer 1 Comments
Point 1: The author should cite the papers with hypoxia (PMID:27027798) and activin A (PMID:29170109) in genetic HO of FOP.
Response 1:
We thank the Reviewer for the comment. The presence of a hypoxic microenvironment is certainly important in sustaining the heterotopic ossification process, therefore, according to the Reviewer’s suggestion we introduce this concept by adding a sentence with the corresponding citation by Wang et al (PMID: 27027798), now indicated as Reference n 68.
Please see Page 5: Lane from 276 to 278.
“This latter is further sustained by the markedly hypoxic microenvironment generated by inflammation in the early FOP lesions, which enhances the BMP signalling and promotes HO formation [68]”
Moreover, given the role of chondrogenesis in the development of HO and the crucial role of Activin A in the contest of FOP, we have introduced a sentence reporting these considerations and the corresponding, suggested citation by Wang et al (PMID:29170109), now indicated as Reference n. 79.
Please see Page 5: Lane from 295 to 298.
“On the contrary, binding of ActA to the mutated receptor carrying FOP-associated variants triggers the downstream Smad1/5/9 signalling [76-78] enhances the endochondral ossification of primary connective tissue progenitor cells of FOP patients [79] thus promoting HO formation”
Point 2: In line 328,"ACVR1Q207D model may not represent the most suitable model to be used in preclinical studies of potential pharmacological treatments for FOP". This model had robust HO than R206H model, shared the same BMP signaling pathway, and used widely for developing some drug candidate. The conclusion is too arbitrary. I suggest to revise the tone.
Interestingly, the author updated a new report of few cases of COVID-19 who developed HO and give a reasonable analysis.
Response 2: We agree with the Reviewer’s comment and in accordance we have modified our sentence, please see Page 6, Lanes from 354 to 357:
“Although ACVR1Q207D shows some functional features different from the naturally occurring ACVR1Q207E and ACVR1R206H mutants, the ACVR1Q207D mouse model presents a robust, BMP-signalling dependent HO formation and extensively used in the preclinical development of inhibitory compounds and drugs [86,89]”
We have also performed a spell check and modified some terms in order to avoid repetitions in the same sentence.
Please note that due to the introduction of new References, citation numbering and list have changed.
Reviewer 2 Report
The authors comprehensively reviewed genetic and acquired heterotopic ossification disorders and summarized the different presentations of heterotopic ossification comparing the human condition and the corresponding animal model. It is well written and provides valuable information to the readers of the journal.
Author Response
Response to Reviewer 2
Comments
The authors comprehensively reviewed genetic and acquired heterotopic ossification disorders and summarized the different presentations of heterotopic ossification comparing the human condition and the corresponding animal model. It is well written and provides valuable information to the readers of the journal.
Response
No revisions required.
Please note that due to the introduction of new References, citation numbering and list have changed.
Reviewer 3 Report
This review manuscript classified mouse models useful for research on ectopic ossification and summarized them according to the cause of onset, and is useful for therapeutic strategies for ectopic ossification and drug deployment.
Regarding the types of ectopic calcification, ectopic calcification associated with higher concentration of calcium or phosphorus in blood is called metastatic calcification, and ectopic calcification without mineral concentration or higher is called dystrophic calcification. It may be associated with chronic kidney disease or hypercalcemia, for example. I think that the description of these distinctions need be described in the introduction.
In the description of constitutively active ALK2, the following paper would be better additionally cited.
Potent inhibition of heterotopic ossification by nuclear retinoic acid receptor-γ agonists Kengo Shimono, Wei-en Tung, Christine Macolino, Amber Hsu-Tsai Chi, Johanna H Didizian, Christina Mundy, Roshantha A Chandraratna, Yuji Mishina, Motomi Enomoto-Iwamoto, Maurizio Pacifici, Masahiro Iwamoto Nature Medicine 2011 https://dx.doi.org/10.1038/nm.2334.
It is well organized as a whole, a minor fix should be enough.
Author Response
Response to Reviewer 3 Comments
Point 1: Regarding the types of ectopic calcification, ectopic calcification associated with higher concentration of calcium or phosphorus in blood is called metastatic calcification, and ectopic calcification without mineral concentration or higher is called dystrophic calcification. It may be associated with chronic kidney disease or hypercalcemia, for example. I think that the description of these distinctions need be described in the introduction.
Response 1:
We thank the Reviewer for this suggestion. We have added two paragraphs in the Introduction section supported by the corresponding citations (see new References from n. 1 to 4), please see Page 1: Lanes from 31 to 42.
“In physiological conditions, bone formation occurs during development through two main pathways: endochondral ossification, in which a cartilage intermediate is progressively replaced by osteogenic cells; and intramembranous ossification, where osteogenic cells differentiate from condensed mesenchymal cells [1,2]. Development of heterotopic bone shares striking molecular and histological features with normal bone development and can be distinguished from ectopic tissue calcification, a pathological event in which different processes lead to the deposition of insoluble calcium salts of variable composition outside bone and teeth, with the involvement of different soft tissues and organs (skin, kidney, lungs, vessels, etc.). Ectopic calcification may be observed in aging or secondary to different pathological conditions in presence of normal calcium and phosphorous values in the plasma (dystrophic calcification). Alternatively, it may be the result of conditions with altered plasma ion levels with precipitation of the exceeding salts (metastatic calcification) [3,4].”
Point 2 In the description of constitutively active ALK2, the following paper would be better additionally cited. Shimono K, et al., Potent inhibition of heterotopic ossification by nuclear retinoic acid receptor-γ agonists. Nat Med. 2011 Apr;17(4):454-60. doi: 10.1038/nm.2334.
Response 2:
We agree with Reviewer's comment and we have accordingly introduced the suggested citation as Reference n. 89, please see Page 6, Lanes from 354 to 357:
“Although ACVR1Q207D shows some functional features different from the naturally occurring ACVR1Q207E and ACVR1R206H mutants, the ACVR1Q207D mouse model presents a robust, BMP-signalling dependent HO formation and extensively used in the preclinical development of inhibitory compounds and drugs [86,89]”
Please note that due to the introduction of new References, citation numbering and list have changed.